# DNA methyltransferase CHROMOMETHYLASE3 prevents *ONSEN* transposon silencing under heat stress

**Kosuke Nozawa**[1]☯, **Jiani Chen**[2]☯, **Jianjun Jiang**[2,3]☯, **Sarah M. Leichter**[2], **Masataka Yamada**[1], **Takamasa Suzuki**[4], **Fengquan Liu**[3], **Hidetaka Ito**[5]*, **Xuehua Zhong**[2]*

1 Graduate School of Life Science, Hokkaido University, Kita10 Nishi8, Kita-ku, Sapporo, Hokkaido, Japan, 2 Wisconsin Institute for Discovery & Laboratory of Genetics, University of Wisconsin-Madison, Madison, Wisconsin, United States of America, 3 Institute of Plant Protection, Jiangsu Academy of Agricultural Sciences, Nanjing, China, 4 College of Bioscience and Biotechnology, Chubu University, 1200 Matsumoto-cho, Kasugai, Aichi, Japan, 5 Faculty of Science, Hokkaido University, Kita10 Nishi8, Kita-ku, Sapporo, Hokkaido, Japan

☯ These authors contributed equally to this work.
* hito@mail.sci.hokudai.ac.jp (HI); xuehua.zhong@wisc.edu (XZ)

**Data Availability Statement:** All RNA-seq data files are available from the DNA Data Bank of Japan (DDBJ) Sequence Read Archive (DRA011115).

## Abstract

DNA methylation plays crucial roles in transposon silencing and genome integrity. CHROMOMETHYLASE3 (CMT3) is a plant-specific DNA methyltransferase responsible for catalyzing DNA methylation at the CHG (H = A, T, C) context. Here, we identified a positive role of CMT3 in heat-induced activation of retrotransposon *ONSEN*. We found that the full transcription of *ONSEN* under heat stress requires CMT3. Interestingly, loss-of-function CMT3 mutation led to increased CHH methylation at *ONSEN*. The CHH methylation is mediated by CMT2, as evidenced by greatly reduced CHH methylation in *cmt2* and *cmt2 cmt3* mutants coupled with increased *ONSEN* transcription. Furthermore, we found more CMT2 binding at *ONSEN* chromatin in *cmt3* compared to wild-type accompanied with an ectopic accumulation of H3K9me2 under heat stress, suggesting a collaborative role of H3K9me2 and CHH methylation in preventing heat-induced *ONSEN* activation. In summary, this study identifies a non-canonical role of CMT3 in preventing transposon silencing and provides new insights into how DNA methyltransferases regulate transcription under stress conditions.

## Author summary

DNA methylation is generally known to silence transposon and maintain genome integrity. Environmental stress has been reported to release the transcriptional silencing of some transposable elements. DNA methylation is involved in the transcriptional restriction of heat-induced *Copia*-type retrotransposon *ONSEN* in Arabidopsis when subjected to heat stress. Here, we identified a non-canonical and positive role of the DNA methyltransferase CMT3 in *ONSEN* reactivation under heat stress. We showed that CMT3

**Funding:** X.Z's. laboratory was supported by the by the NSF awards (MCB-1552455 and MCB-2043544), the NIH-MIRA (R35GM124806), and the USDA & National Institute of Food and Agriculture grant (Hatch 1012915). H.I.'s laboratory was supported by the Grant-in-Aid for Scientific Research on Innovative Areas (JP15K21750), Grant-in-Aid for JSPS Fellows (18K06050), the Grant-in-Aid for Scientific Research C (18J11120), and Hokkaido University President's Encouragement Program for Sending Young Faculty Members Abroadand the Grant-in-Aid for Scientific Research C (18K06050). The funders had no role in study design, data collection and analysis, decision to publish, or preparation of the manuscript.

**Competing interests:** The authors have declared that no competing interests exist.

prevents CMT2-mediated CHH methylation and H3K9me2 accumulation under heat at *ONSEN* chromatin to modulate *ONSEN* transcription. Our work revealed the molecular mechanism of CMT3 in heat-induced *ONSEN* activation and sheds new light on the survival mechanism of certain transposons in the host genome under stress conditions.

## Introduction

DNA methylation is vital for silencing of repetitive sequences and transposable elements (TEs) to ensure genome stability and integrity [1]. Plant DNA methyltransferases methylate cytosine in three different contexts, including CG, CHG, and CHH (H = A, C, or T) [2]. In *Arabidopsis thaliana*, the *de novo* DNA methylation is mediated by DOMAINS REARRANGED METHYL-TRANSFERASE 2 (DRM2) through the RNA-directed DNA methylation (RdDM) pathway. This process involves the biogenesis of small interfering RNA (siRNA) and the recruitment of DRM2 to target DNA sequences for methylation [2]. The maintenance of CHG and CHH methylation is primarily mediated by CHROMOMETHYLASE3 (CMT3) and CHROMO-METHYLASE2 (CMT2), respectively, through a self-reinforcing mechanism [3,4]. Specifically, CMT2 and CMT3 methylate DNA through the dual binding of H3K9 di-methylation (H3K9me2)-containing nucleosomes via their bromo adjacent homology (BAH) and Chromo domains [3]. In turn, histone methyltransferases SUVH4/KYP, SUVH5, and SUVH6 bind CMT3-methylated CHG or CMT2-methylated CHH and deposit two methyl groups on H3K9 (H3K9me2) [4,5]. CMT3 also tends to methylate CHG at large TEs and leads to the transcriptional repression of these TEs [4,6]. CMT2 catalyzes CHH methylation predominantly at large TEs localized to heterochromatin [7]. In contrast, DRM2 maintains CHH methylation at short TEs, the border of large TEs, and other repeat sequences in euchromatic regions [2,7].

TEs are widely distributed in the eukaryotic genome, ranging from 15% TEs in the *Arabidopsis thaliana* genome to 85% in the *Zea mays* genome [8,9]. There are two classes of TEs, DNA transposons that transpose by a 'cut-and-paste' mechanism, and retrotransposons that mobilize by a 'copy-and-paste' mechanism through an RNA intermediate [8]. *ONSEN/COPIA78* is a Ty1/*copia*-like retrotransposon with long terminal repeat (LTR) at both ends in *Arabidopsis thaliana* [10]. Because the LTRs at the 5' and 3' ends are identical at the time of insertion, the sequence differences between the LTR pairs can be used to infer the timing of the transpositions. *ONSEN* contains eight copies (*ONSEN1-8*) in the genome of the *Arabidopsis thaliana* Columbia (Col-0) ecotype [11]. Three of them (*ONSEN1-3*) have identical 5' and 3' LTR sequences, suggesting that they are recent insertions [12]. We previously found that *ONSEN* preferentially targets euchromatic regions and affects gene expression [13]. For instance, a new *ONSEN* insertion at an abscisic acid (ABA) responsive gene *ABI5* leads to an ABA-insensitive phenotype [13].

Although most TEs are epigenetically silenced by DNA methylation, environmental stresses can temporarily release TEs and heterochromatin silencing [14,15]. Previous studies have reported that *ONSEN* is activated by heat stress in Arabidopsis [12]. Under heat stress conditions, *ONSEN* is transcriptionally activated by the heat-responsive transcription factors including HSFA2 by recognizing the heat shock elements located within the LTR of *ONSEN*, further resulting in the production of extrachromosomal DNA [11]. Heat-induced *ONSEN* accumulation was significantly increased in the mutant of NRPD1, a subunit of RNA polymerase IV (Pol IV) and a key factor in siRNA biosynthesis and RdDM pathway, suggesting that the siRNA pathways are involved in the repression of *ONSEN* transcription under heat stress [12]. However, the molecular mechanism of DNA methylation in *ONSEN* reactivation under heat stress remains unknown.

Here, we investigated *ONSEN* transcription in DNA methyltransferase mutants including *drm1 drm2*, *cmt2*, and *cmt3*. Surprisingly, we found that heat-induced transcription of *ONSEN* is dampened in *cmt3* mutant under heat stress. Loss-of-function of CMT3 results in an increased CHH methylation at *ONSEN*. This methylation is mediated by CMT2, as knocking out CMT2 in *cmt3* mutant leads to reduced CHH methylation accompanied with *ONSEN* accumulation. This phenomenon of reduced TE expression in the *cmt3* mutant is distinct from the previously reported findings that CMT3 mediates transposon silencing. We also found increased CMT2 binding and H3K9me2 level after heat stress at the *ONSEN* locus in the *cmt3* mutant. Furthermore, we found reduced CMT2 protein level and H3K9me2 level at *ONSEN* under heat treatment. Together, this study identifies a non-canonical role of CMT3 in preventing TE silencing and provides new insights into the regulatory mechanism of DNA methylation on transposon reactivation under environmental stress.

## Results

### *ONSEN* transcriptional activation is dampened in *cmt3* under heat stress

We have previously shown that heat stress activates *ONSEN* transcription, and the heat–induced *ONSEN* transcripts are further accumulated in mutants impaired in the siRNA bio-genesis pathway [12]. SiRNAs guide DNA methylation at target loci containing homologous sequences through the RdDM pathway, leading to transcriptional silencing [2]. To determine the role of DNA methylation in *ONSEN* transcription under heat stress, we examined its transcript level in DNA methylation-deficient mutants by quantitative reverse transcription PCR (RT-qPCR). Consistent with previously published results [12], higher *ONSEN* RNA level was noted in *drm1 drm2* (*drm1/2*) and *cmt2* mutants compared to Col-0 under heat stress (S1 Fig). Surprisingly, *ONSEN* transcripts decreased more than 50% in *cmt3-11* (hereafter called *cmt3*) mutants compared to Col-0 under heat stress (Figs 1A and S1). Similar results were observed in *cmt3-7* mutants of the Landsberg *erecta* (Ler) accession (Fig 1A). These data suggest that CMT3 plays an important role in the full transcription of *ONSEN* under heat stress.

To explore whether CMT3 negatively regulates other transposon families under heat stress, we performed transcriptome analysis using high-throughput RNA-seq of the *cmt3* mutant and Col-0 under non-stress and heat stress conditions (S1 Table and S1 Data). However, we did not find any other transposons that were down regulated in *cmt3* compared with Col-0 under heat stress. *ONSEN* was the only heat-induced transposon family that showed reduced transcript levels in *cmt3* under heat stress (Fig 1B and 1C). To further investigate whether CMT3 regulates additional heat-induced transposons, we examined the transcript level of two prominent heat-responsive *COPIA* families, *ROMANIAT5* and *AtCOPIA37* [10], in *cmt3* mutants under heat stress. The transcript level of *AtCOPIA37* was unchanged and the expression of *ROMANIAT5* was only slightly reduced in *cmt3* (S2 Fig). Together, these results suggest that CMT3-dependen transposon activation under heat stress is mostly likely *ONSEN*-specific.

### CMT3 regulates *ONSEN* transcription independent of the heat-shock response pathway

Heat shock factors and heat shock proteins are key players in response to heat stress [16–19]. Heat-induced transcription factor HSFA2 is required for the transcriptional activation of *ONSEN* during heat stress [11,20]. It prompted us to study whether the dampened *ONSEN* activation in *cmt3* resulted from a dysfunctional heat-shock response pathway. We examined the transcript level of two heat-responsive genes, *HSFA2* and *HSP90.1* [21], in *cmt3* and Col-0. Expression of both *HSFA2* and *HSP90.1* was unchanged in *cmt3* under heat stress relative to

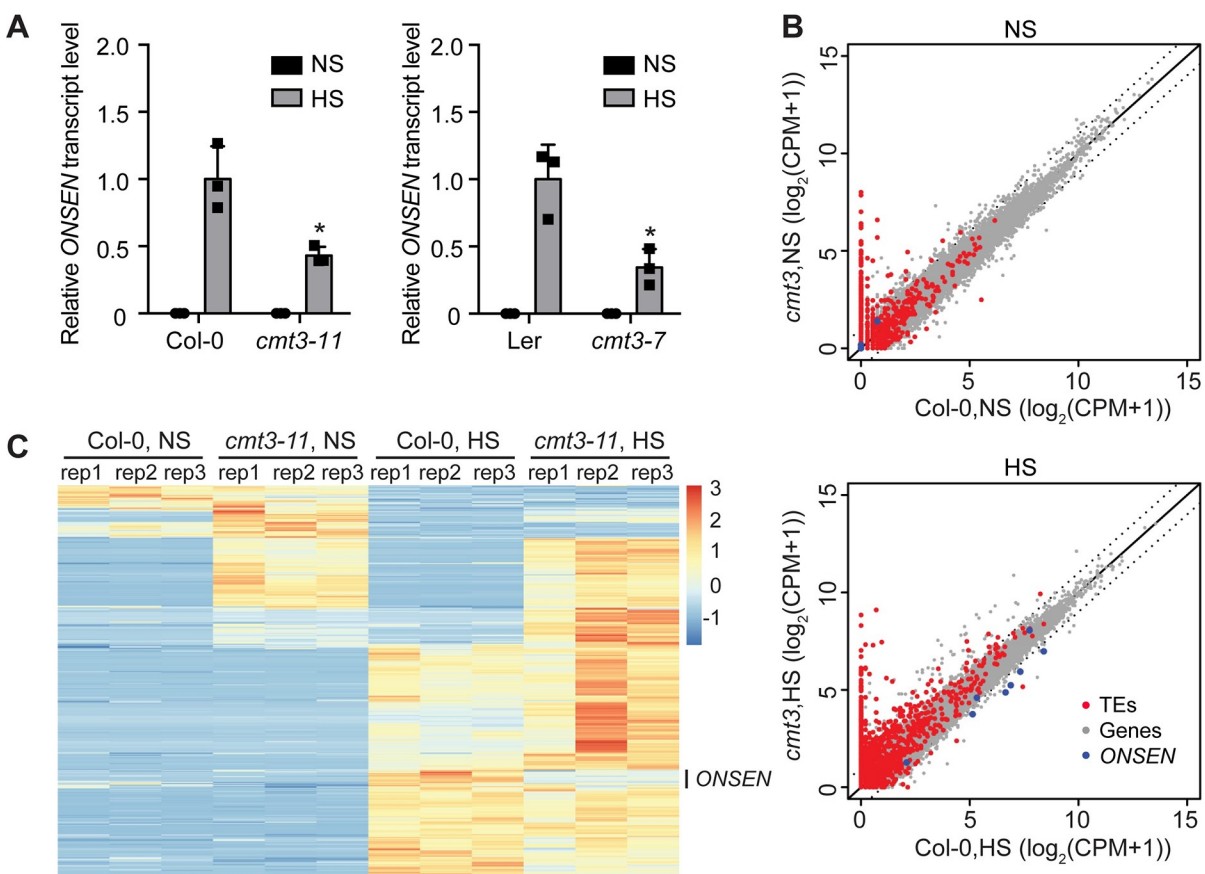

**Fig 1. Heat-induced *ONSEN* transcript accumulation is dampened in *cmt3* mutants.** (A) RT-qPCR of *ONSEN* transcripts in *cmt3* mutants from Columbia-0 (*cmt3-11*, left) and Landsberg *erecta* (*cmt3-7*, right) backgrounds under non-stress (NS) and heat stress (HS). The *ONSEN* transcripts were normalized to *18S rRNA*. All bars represent mean + SD from three biological replicates. Student's *t*-test. *P < 0.05. (B) The scatterplots of count per million reads (CPM) of protein-coding genes and transposable elements (TEs) in *cmt3* versus Col-0 under NS (top) and HS (bottom). Dash lines indicate two-fold changes. (C) Heat maps of differentially expressed TEs in Col-0 and *cmt3-11* under NS and HS from three biological replicates (rep1 to rep3). Color bar indicates the Z-score.

Col-0 (S3 Fig), suggesting that CMT3 regulates *ONSEN* transcription independent of the heat shock response pathway.

It has been reported that prolonged heat stress can lead to decondensation of chromatin [22,23]. Therefore, the reduced *ONSEN* transcription in *cmt3* may likely result from diminished chromatin decondensation after heat stress. To test this idea, we performed 4′,6-diamidino-2-phenylindole (DAPI) staining, a fluorescent stain of nuclear DNA, to visualize the chromatin compactness. Under heat stress, about 80% of the nuclei were dispersed in Col-0 (S4 Fig), confirming that heat induced chromatin decondensation. In *cmt3*, a similar percentage of nuclei were dispersed compared to Col-0 under heat stress (S4 Fig), indicating that the reduced *ONSEN* transcription in *cmt3* is less likely caused by a more compacted chromatin state.

## BAH, chromo, and catalytic domains of CMT3 are required for the activation of *ONSEN*

CMT3 is a DNA methyltransferase mediating CHG methylation, which depends on recognizing histone H3K9me2 via both BAH and chromo domains [3]. To investigate the importance of BAH and chromo domains in CMT3-mediated *ONSEN* regulation, we used the published

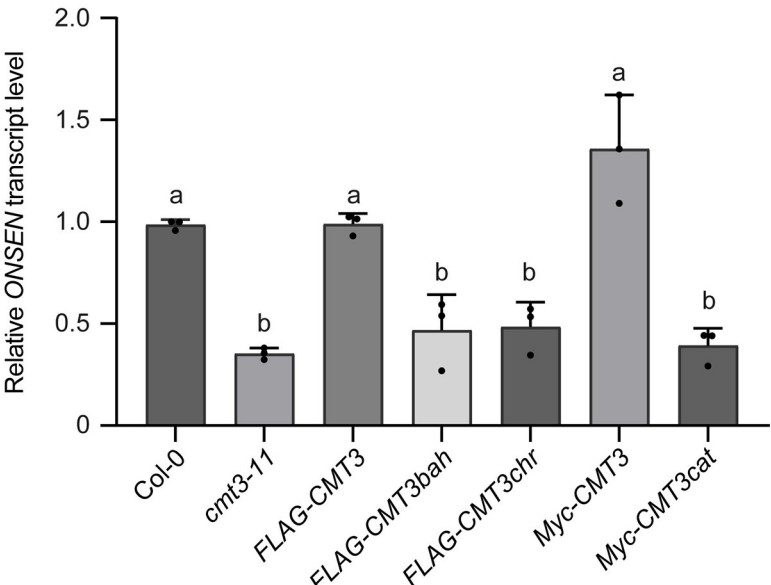

**Fig 2. BAH, chromo, and catalytic domains of CMT3 are required for heat-induced *ONSEN* transcription.** Bar graph showing relative *ONSEN* transcript level under heat stress in Col-0, *cmt3-11*, and transgenic plants expressing wild-type CMT3 (*FLAG-CMT3*), BAH domain mutant (*FLAG-CMT3bah*), chromo domain mutant (*FLAG-CMT3chr*), wild-type CMT3 (*Myc-CMT3*), and catalytic domain mutant (*Myc-CMT3cat*) in *cmt3-11* background. The relative transcripts were first normalized to *ACT7*, and then Col-0. All bars represent mean + SD from three biological replicates. Different letters represent significant differences (P < 0.05 by Student's *t*-test) between samples.

FLAG-tagged CMT3 transgenic plants carrying mutations within the BAH (FLAG-CMT3bah) and chromo domain (FLAG-CMT3chr) that are defective in H3K9me2 binding [3]. We next heat treated these plants and found that wild-type CMT3 (FLAG-CMT3) completely restored the down regulation of *ONSEN* in *cmt3* under heat stress. In contrast, mutations within the BAH or chromo domains failed to recover the reduced *ONSEN* transcription in *cmt3* (Fig 2). This result suggests that a functional H3K9me2 binding activity of CMT3 is required for a full transcriptional response of *ONSEN* under heat stress.

Since BAH and chromo domains are required for CMT3 in maintaining the DNA methylation through the self-reinforcing CHG-H3K9 methylation loop, we investigated whether the CMT3-mediated DNA methylation is indispensable for *ONSEN* activation under heat stress. To address this question, we heat treated the published Myc-tagged CMT3 transgenic lines expressing either wild-type CMT3 (Myc-CMT3) or catalytically inactive CMT3 mutant (Myc-CMT3cat) in *cmt3* background [3]. We found that the reduced *ONSEN* transcripts in *cmt3* were restored in the wild-type CMT3 (Myc-CMT3), but not in the CMT3 catalytic mutant (Fig 2), suggesting that catalytic activity of CMT3 is required for the full activation of *ONSEN* under heat stress.

## CHH hypermethylation at *ONSEN* in *cmt3* is mediated by CMT2

Given the importance of CMT3-mediated DNA methylation in heat-induced *ONSEN* activation, we examined the DNA methylation levels at the *ONSEN* locus using previously published bisulfite-sequencing (BS-seq) data [6]. We found a low number of methylated CHG sites at the *ONSEN* locus (*ONSEN2*, *AT3G61330*) in Col-0 (Fig 3A). Interestingly, CHH methylation was abundant across the *ONSEN* locus (Fig 3A), consistent with the findings that the LTR of *ONSEN* only contains CHH methylation [11]. Furthermore, the CHH methylation level

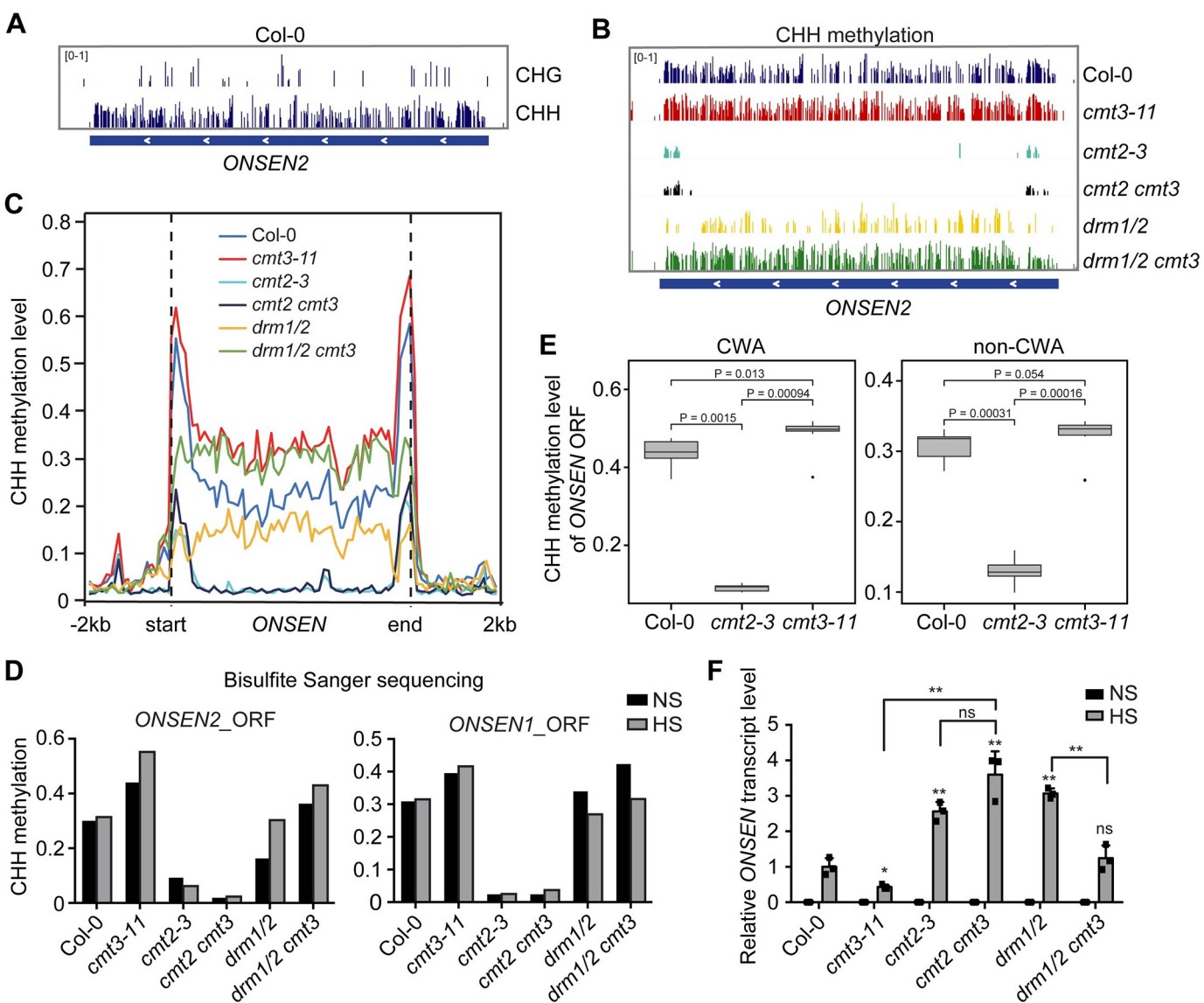

**Fig 3. CHH hypermethylation at *ONSEN* in *cmt3* is mediated by CMT2.** (A) Genome browser view of CHG and CHH methylation within *ONSEN2* (AT3TE92525/AT3G61330), in Col-0. (B) Genome browser view of CHH methylation within *ONSEN2* in Col-0, *cmt3-11*, *cmt2-3*, *cmt2 cmt3*, *drm1/2*, and *drm1/2 cmt3*. (C) Metaplots showing average CHH methylation of 8 *ONSEN* copies in the corresponding mutants. -2kb indicates the upstream 2000 bp of the start site, and 2kb indicates the downstream of 2000 bp of the end site. (D) Bisulfite Sanger-sequencing for CHH methylation level of the ORFs of *ONSEN2* and *ONSEN1*, two *ONSEN* copies highly induced by heat stress, in the corresponding mutants under non-stress (NS) and heat stress (HS). (E) Boxplots showing CWA (CAA and CTA) and non-CWA CHH methylation level in Col-0, *cmt2-3*, and *cmt3-11* of the open reading frame (ORF) of all eight copies of *ONSEN*. P-values are from Wilcoxon test. (F) RT-qPCR of *ONSEN* transcripts in the corresponding mutants under NS and HS. All bars represent mean + SD from two biological replicates. The *ONSEN* transcripts were normalized to *18S rRNA*. Student's *t*-test, *P < 0.05; **P < 0.01. ns indicates not significant.

showed a great increase in *cmt3* compared to Col-0, particularly in the body region of *ONSEN* (Fig 3B and 3C). Besides *ONSEN*, we identified additional TEs with CHH hypermethylation in *cmt3* (S5A Fig and S2 Data). Further analysis showed that these TEs are enriched in TE families such as DNA and DNA/MuDR and are featured with significantly less CHG content, but greater CHH content compared to all TEs across the genome (S5B and S5C Fig). Bisulfite Sanger-sequencing of the ORFs of *ONSEN1* (*AT1G11265*) and *ONSEN2* also displayed an increase of CHH methylation in *cmt3* (Fig 3D). Taken together, these data indicate that CMT3-regulated *ONSEN* transcription is highly correlated with CHH methylation.

CHH methylation of Arabidopsis TEs is maintained by both RdDM-dependent DRM2 and RdDM-independent CMT2 methyltransferases [4]. DRM2 tends to methylate euchromatic TEs and edges of large TEs, while CMT2 methylates TEs located at pericentromeric heterochromatin [4,7]. Next, we determined whether the increased CHH methylation at *ONSEN* in *cmt3* is mediated by DRM2 or CMT2. The published BS-seq data of *cmt2*, *cmt2 cmt3*, *drm1/2* and *drm1/2 cmt3* was analyzed to examine the CHH methylation level at *ONSEN* [6]. We found that CHH methylation was reduced at the edges of the *ONSEN* in the *drm1/2* double mutant, but was almost completely lost in *cmt2*, particularly in the TE body (Fig 3C). Knocking out CMT2 in *cmt3* (*cmt2 cmt3*) rendered a complete loss of CHH methylation, whereas loss of DRM1 and DRM2 in *cmt3 (drm1/2 cmt3)* did not cause a change at the TE body (Fig 3B–3D). It is reported that CMT2 preferentially catalyzes CWA (i.e. CTA or CAA) methylation in Arabidopsis [24]. We then analyzed the CWA and non-CWA CHH methylation levels over the open reading frames (ORFs) of all 8 copies of *ONSEN* and found that CWA methylation level was significantly higher (P = 0.013) in *cmt3* compared with that of Col-0 (Fig 3E). Together, these data suggest that CMT2 is the primary DNA methyltransferase responsible for CHH methylation at the *ONSEN* locus.

To assess whether the change of the CHH methylation level is associated with a differential transcription, we examined *ONSEN* transcripts in *cmt2*, *cmt2 cmt3*, *drm1/2*, and *drm1/2 cmt3* under heat stress by RT-qPCR. *ONSEN* transcripts were significantly increased in *cmt2* (Figs 3F and S6), consistent with the decreased CHH methylation level. Knocking out CMT2 in *cmt3* restored the reduced *ONSEN* transcripts in *cmt3* to a level similar to *cmt2*, whereas *drm1/2 cmt3* showed a slightly higher *ONSEN* level but was still significantly lower than the *drm1/2* mutant (Figs 3F and S1 and S6). Taken together, these results demonstrate that the dampened *ONSEN* activation in *cmt3* under heat stress is largely due to the increased CHH methylation mediated by CMT2.

## CMT3 regulates CMT2 binding to *ONSEN* chromatin

To understand the molecular mechanism underlying the regulation of CMT3 on CMT2-mediated CHH methylation at the *ONSEN* locus, we first explored whether the increased CHH methylation at *ONSEN* in the *cmt3* mutant is due to an increase of CMT2 at the transcriptional level. We found that the *CMT2* expression level was unchanged in *cmt3* relative to Col-0 (S7A Fig), suggesting that CHH hypermethylation at *ONSEN* in *cmt3* is unlikely due to the increased *CMT2* expression.

The similarity of CMT2 and CMT3 in their H3K9me2 binding capacity prompted us to hypothesize that CMT3 may compete with CMT2 for H3K9me2 binding on the *ONSEN* locus. To test this hypothesis, we first examined the enrichment of CMT3 at the *ONSEN* locus by performing chromatin immunoprecipitation coupled with quantitative PCR (ChIP-qPCR) using previously published FLAG-tagged CMT3 lines in *cmt3* [3]. We found that CMT3 is indeed enriched at *ONSEN* (Fig 4A and 4B). Next, we determined whether CMT2 is also enriched at *ONSEN* locus by generation of epitope-tagged CMT2 transgenic plants expressing full-length genomic CMT2 fused with FLAG tag in *cmt2* mutant background (CMT2-FLAG/*cmt2*). We performed a similar ChIP-qPCR and found the enrichment of CMT2 at *ONSEN* (Fig 4C) under normal conditions. To further investigate the impact of CMT3 on CMT2 binding at *ONSEN*, we crossed CMT2-FLAG/*cmt2* into *cmt3* mutant to generate homozygous CMT2-FLAG in *cmt2 cmt3* background (CMT2-FLAG/*cc*) lines. The CMT2 protein level was not affected by CMT3 mutation (S7B Fig). We subsequently performed ChIP-qPCR and found that CMT2 binding at *ONSEN* was increased in CMT2-FLAG/*cc* compared to

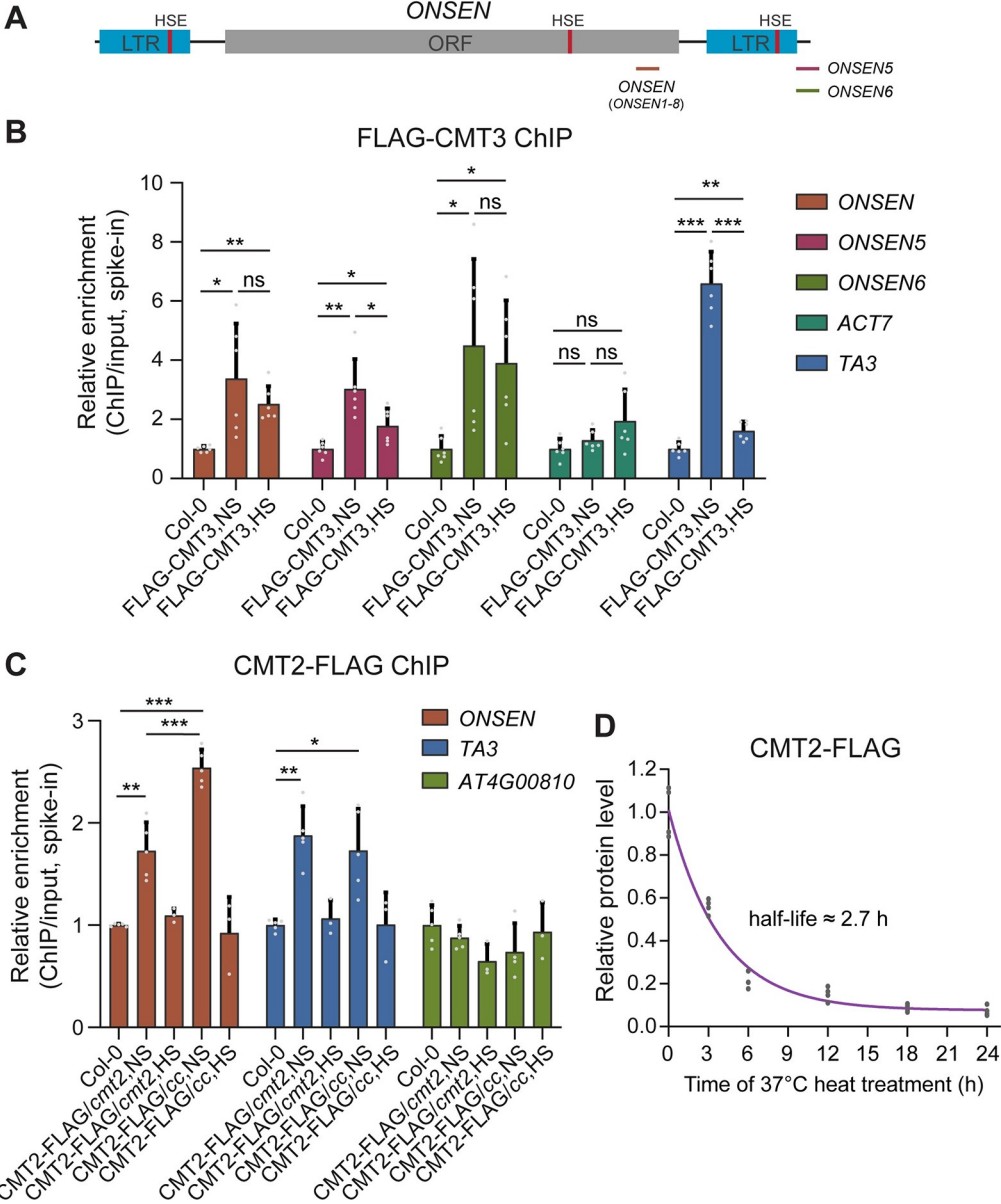

**Fig 4. CMT2 binding on *ONSEN* is increased in *cmt3* mutant.** (A) A diagram showing *ONSEN* structure and the locations of ChIP-qPCR amplicons. LTR, long terminal repeat; ORF, open reading frame; HSE, heat-shock element. (B) ChIP-qPCR showing CMT3 enrichment at *ONSEN* in Col-0 and CMT3 tagged line in *cmt3-11* (FLAG-CMT3) under non-stress (NS) or heat stress (HS) conditions. After normalization to input, ChIP samples were further normalized to the respective spike-in of human chromatin. *TA3* was used as a positive control, and *ACT7* was used as a negative control. All bars represent mean + SD from two biological replicates with all technical replicates shown. Student's *t*-test, *P < 0.05; **P < 0.01; ***P < 0.001. ns indicates not significant. (C) ChIP-qPCR showing CMT2 enrichment at *ONSEN* in Col-0 and CMT2 tagged line in *cmt2* (CMT2-FLAG/*cmt2*) and *cmt2 cmt3* (CMT2-FLAG/*cc*) under NS or HS conditions. After normalization to input, ChIP samples were further normalized to the respective spike-in of human chromatin. *AT4G00810* was used as a negative control. All bars represent mean + SD from two (NS) or one (HS) biological replicates with all technical replicates shown. Student's *t*-test, *P < 0.05; **P < 0.01; ***P < 0.001. ns indicates not significant. (D) CMT2 protein level in a time-course of heat (37°C) treatment. The fitting curve was generated using a one-phase decay model, and a half-life was calculated.

CMT2-FLAG/*cmt2* (Fig 4C), but not on the negative control *AT4G00810* [25]. This result suggests that the presence of CMT3 leads to less occupancy of CMT2 at *ONSEN* chromatin.

We next investigated the effect of heat stress on the binding of CMT3 and CMT2 at *ONSEN* locus. As heat stress will promote the generation of extrachromosomal DNA [12], we used primers located in both *ONSEN* ORF and primers flanking the borders for ChIP-qPCR (Fig 4A). After heat stress, we noticed significant CMT3 binding at *ONSEN* locus, though the enrichment level seems to be reduced (Fig 4B). However, we could not detect CMT2 binding in both CMT2-FLAG/*cmt2* and CMT2-FLAG/*cc* lines after heat stress (Fig 4C). Next, we examined CMT2 protein levels in these *CMT2-FLAG* transgenic plants upon a series of heat treatment time points. Surprisingly, CMT2 proteins degraded quickly when plants were subjected to heat treatment and the calculated half-life is about 2.7 hours (Figs 4D and S8A and S8B). There were no noticeable changes in both CMT3 and DRM2 protein levels after heat stress (S8C and S8D Fig). Furthermore, we examined the expression level of CMT2 upon heat stress and found that CMT2 transcript level was reduced at the 3 hour time-point and then remained stable with prolonged heat treatment (S8E Fig).

## Ectopic accumulation of H3K9me2 at *ONSEN* under heat stress in *cmt3* mutant

CMT2-mediated CHH methylation depends on H3K9me2 [26]. To determine whether increased CMT2 binding to *ONSEN* in *cmt3* is correlated with H3K9me2 abundance, we examined the H3K9me2 level at *ONSEN* using previously published data [4]. We found that H3K9me2 is abundant at *ONSEN* in Col-0 but is greatly reduced in the loss-of-function H3K9 methyltransferase mutant of SUVH4, SUVH5, and SUVH6 (*suvh456*) (Fig 5A). Furthermore, the *suvh456* mutant showed much lower CHH methylation compared to Col-0 in the body region of *ONSEN* (Fig 5B and 5C). We further analyzed the CWA and non-CWA CHH methylation of the ORFs of all 8 copies of *ONSEN* and found that CWA methylation level was significantly increased in *cmt3* compared with that of wild-type and *suvh456* (Fig 5D). We also found more *ONSEN* transcripts in *suvh456* than Col-0 after heat stress (Fig 5E), similar to that of *cmt2* (Fig 3F). Knocking out SUVH4, SUVH5, and SUVH6 in the *cmt3* mutant restored *ONSEN* transcription to a similar level to *suvh456* (Fig 5E). These results suggest that H3K9me2 plays an important role in the CMT3-mediated regulation of *ONSEN* under heat stress.

Next, we performed H3K9me2 and H3 ChIP-qPCR in *cmt3*, *cmt2* and Col-0 to compare the H3K9me2 abundance under non-stress and heat stress. Under non-stress, we found a significantly lower H3K9me2 level at *ONSEN* in *cmt2* compared to Col-0 (Fig 5F). Under heat stress, H3K9me2 level is significantly reduced at both *ONSEN* and *TA3* locus in Col-0 (Figs 5F and S9A and S9B). This H3K9me2 reduction is correlated with the decondensation of chromatin caused by heat stress since H3K9me2 is enriched in the heterochromatic region [22]. It is also consistent with previous reports showing H3K9me2 reduction at *ONSEN* after long-term heat stress (37°C for 30 h) [27], and the upregulation of many TEs under heat stress in both Col-0 and *cmt3* (Fig 1C, [10,28]). Surprisingly, we found that the H3K9me2 level at *ONSEN* is significantly higher in *cmt3* mutant than that in Col-0 under HS (Figs 5F and S9A), despite their similar global H3K9me2 levels (S9C and S9D Fig). Consistently, we found that the *ONSEN* transcriptional activation is much slower in *cmt3* compared with Col-0, while the *ONSEN* transcription activation was much faster in *cmt2* (S10 Fig). Taken together, H3K9me2 and CHH methylation function collaboratively in regulating heat induced *ONSEN* transcription.

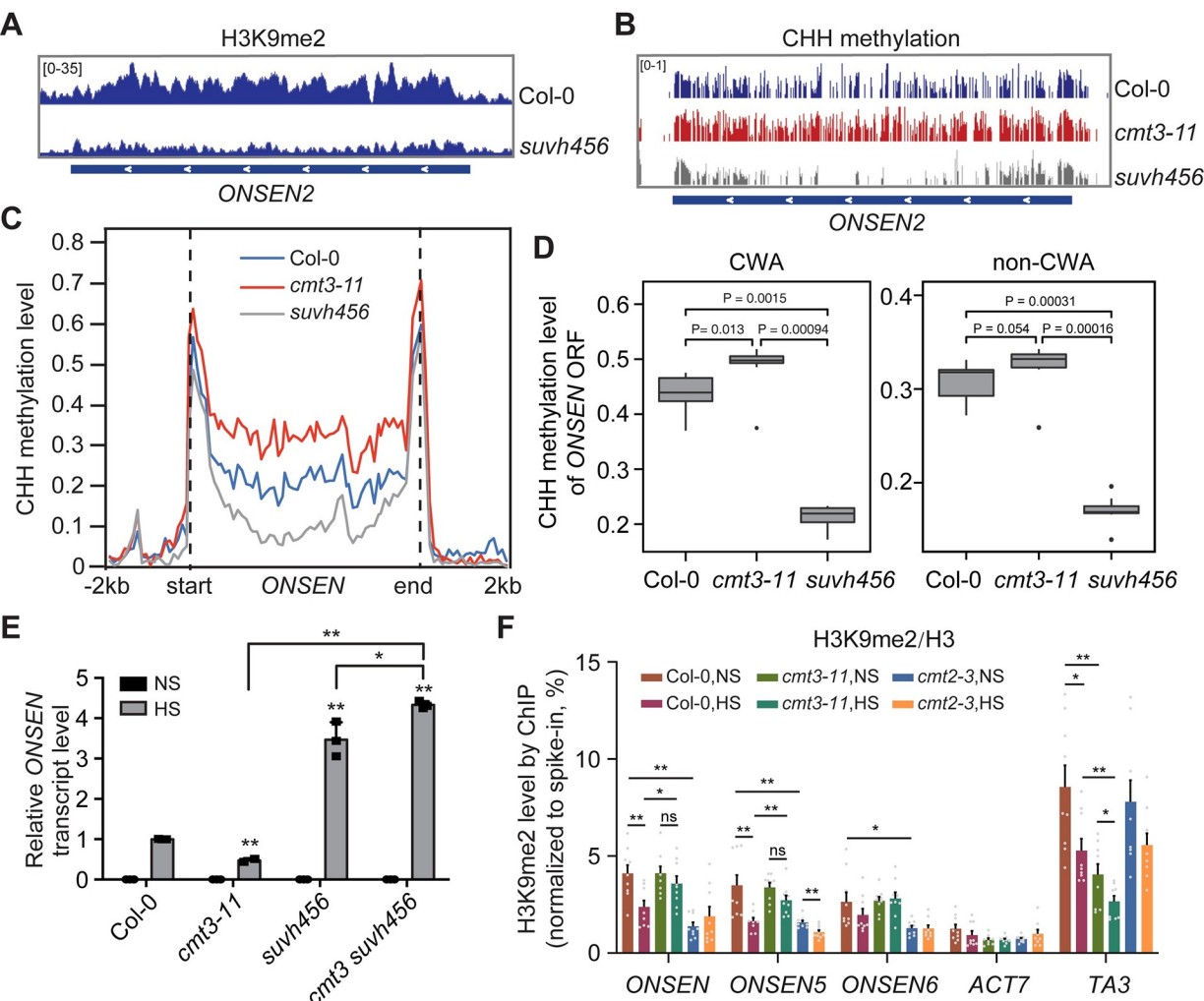

**Fig 5. CMT3-mediated *ONSEN* activation is regulated by H3K9 methylation.** (A) Genome browser view of H3K9me2 abundance at *ONSEN* in Col-0 and the SUVH4, SUVH5, and SUVH6 loss-of-function mutant (*suvh456*). (B) Genome browser view of CHH methylation at *ONSEN2* in Col-0, *cmt3-11*, and *suvh456*. (C) Metaplots showing average CHH methylation of 8 *ONSEN* copies. -2kb indicates the upstream 2000 bp of the start site, and 2kb indicates the downstream of 2000 bp of the end site. (D) Boxplots showing CWA (CAA and CTA) and non-CWA CHH methylation level in Col-0, *cmt3-11*, and *suvh456* of the open reading frame (ORF) of all eight copies of *ONSEN*. P-values are from Wilcoxon test. (E) Relative *ONSEN* transcript level in the corresponding mutants under non-stress (NS) and heat stress (HS). The *ONSEN* transcripts were normalized to *18S rRNA*. All bars represent mean + SD from three biological replicates. Student's *t*-test. *P < 0.05, **P < 0.01. (F) ChIP-qPCR showing H3K9me2 level at *ONSEN* in Col-0,*cmt3*, and *cmt2* under NS and HS. H3K9me2 ChIP samples were first normalized to parallel H3 ChIP, and then to the respective spike-in human chromatin. *ACT7* served as a negative control. *TA3* was used as a positive control. All bars represent mean + SEM from three biological replicates with all technical replicates shown. Student's *t*-test, *P < 0.05; **P < 0.01; ***P < 0.001. ns indicates not significant.

## Discussion

DNA methylation is known to suppress transposon activation to maintain genome integrity in both plants and animals. In this study, we found that a plant-specific DNA methyltransferase CMT3 plays a positive role in the transcriptional activation of a unique transposon family under heat stress. We propose a model wherein CMT3 modulates heat-induced *ONSEN* transcriptional activation by regulating both H3K9me2 and CMT2-mediated CHH methylation. Under non-stress conditions, loss of CMT3 leads to increased CMT2-mediated CHH methylation. Under heat stress, the CHH methylation and H3K9me2 remain high at the *ONSEN* locus

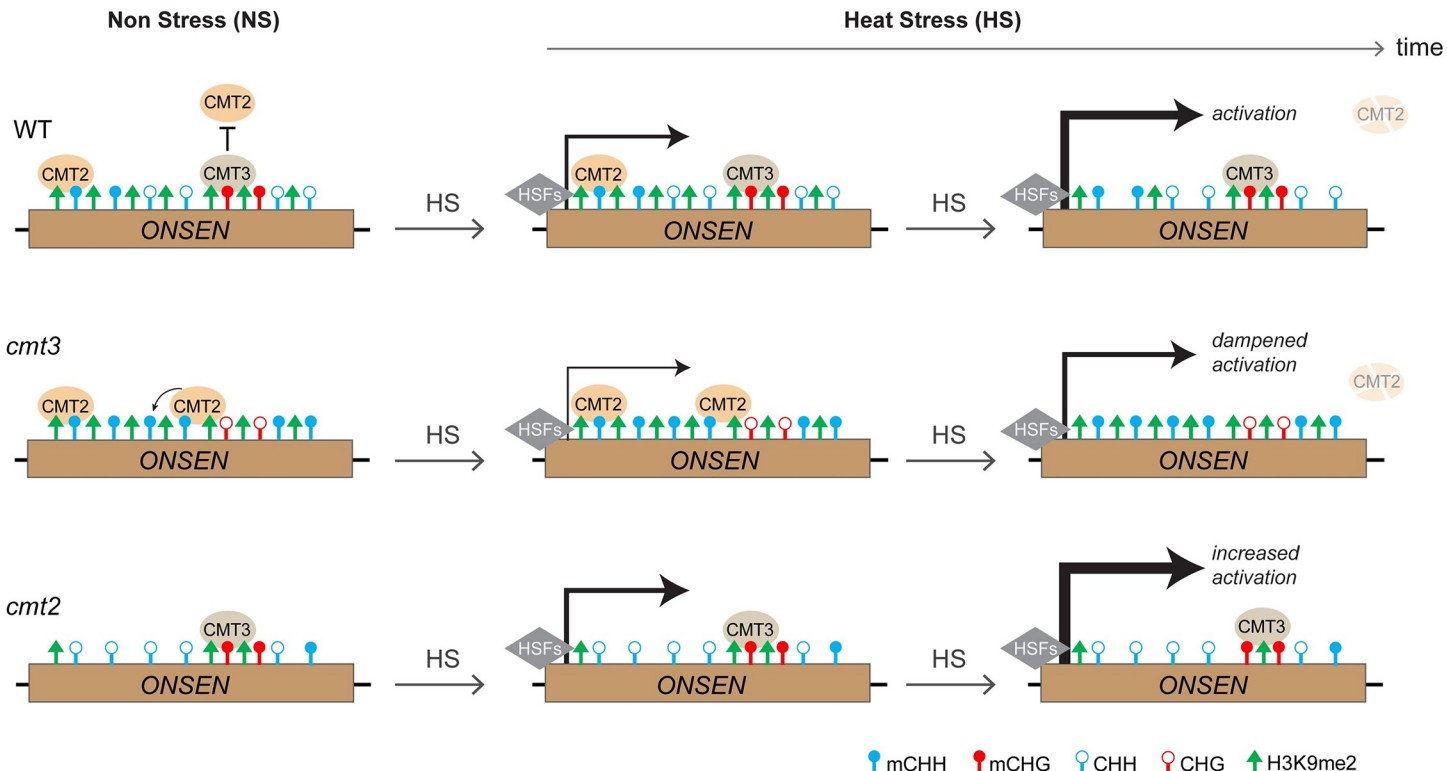

**Fig 6. Working model for CMT3 function in heat-induced *ONSEN* activation.** Under non-stress (NS) conditions, CMT2 and CMT3 are recruited to *ONSEN* locus by H3K9me2 via a self-reinforcing loop. In the wild-type (WT), CMT3 in the *ONSEN* region suppresses the binding of CMT2. Loss-of-function CMT3 induces increased CHH methylation mediated by CMT2. Upon heat stress (HS), *ONSEN* transcription is activated by heat shock factors (HSFs). The high levels of CHH methylation and H3K9me2 at *ONSEN* chromatin in *cmt3* dampen its transcriptional activation by HSFs. In *cmt2*, the CHH methylation and H3K9me2 levels are low, leading to increased *ONSEN* transcription under heat stress. CMT2 protein is degraded by prolonged heat stress.

in *cmt3* mutant, in contrast to the reduced H3K9me2 in wild type, leading to dampened *ONSEN* transcriptional activation in *cmt3* (Fig 6). The CHH methylation and H3K9me2 levels are low in *cmt2* mutant, resulting in increased *ONSEN* activation by heat stress (Fig 6).

CMT3 is responsible for CHG methylation and transposon silencing, while its paralog CMT2 is mainly responsible for CHH methylation jointly with DRM2 [3,7]. Here, we reveal a non-canonical role of CMT3 in preventing retrotransposon *ONSEN* silencing under heat stress. We provide several lines of evidence that CMT3-regulated *ONSEN* transcription under heat stress is mainly through CMT2-mediated CHH methylation. First, the CHH methylation at *ONSEN* was primarily mediated by CMT2 (Fig 3B–3E). The CHH methylation was depleted in the *cmt2* mutant but not in the *drm1 drm2* mutant, particularly within the TE body (Fig 3B–3E). Similar evidence showed a dramatic reduction of CHH methylation at *ONSEN* in *cmt2 cmt3*, but not in the *drm1 drm2 cmt3* triple mutant, further confirming that CMT2 catalyzes CHH methylation at *ONSEN*. Second, we show that the CHH methylation level was negatively correlated with *ONSEN* transcription (Fig 3F). *ONSEN* was downregulated in the *cmt3* mutant, but was significantly upregulated when knocking out CMT2, as shown in the *cmt2 cmt3* double mutant. It is noted that the *ONSEN* transcript was slightly increased in the *drm1/ 2 cmt3* mutant, which may be due to the reduced CHH methylation at the LTR regions. It is possible that the RdDM pathway regulates the *ONSEN* CHH methylation at LTR regions while CMT2 contributes to the TE body methylation. This is consistent with a previous report indicating that CMT2 is required for TE body methylation on long TEs, including the *Copia* family

[7]. Consistently, a recent study also found that DNA methylation was required for histone H1-mediated repression of *ONSEN* in heat stress response [29]. It has been demonstrated that a DNA methyltransferase inhibitor blocked the up-regulation of *ONSEN* in the heat-treated *h1* mutant [29]. Coinciding with our finding, the *ONSEN/COPIA78* was upregulated in heat-treated *cmt2* mutants [29]. Additionally, it has been reported that CHH methylation variation in the accessions of *Arabidopsis thaliana* grown at two different temperatures was associated with natural CMT2 variation [30]. Our study presented here exemplifies the role of CMT2-mediated CHH methylation in CMT3-regulated *ONSEN* activation under heat stress.

Both CMT2 and CMT3 can bind to H3K9me2-containing nucleosomes through the dual binding of the BAH domain and chromodomain, leading to CHH and CHG methylation, respectively [3,4]. Here, our results suggest that CMT2 and CMT3 may compete for H3K9me2 binding at the *ONSEN* locus. We demonstrated that both CMT3 and CMT2 were enriched at the *ONSEN* chromatin (Fig 4B and 4C). The binding of CMT3 on H3K9me2 may restrict the amount of H3K9me2 that CMT2 can access, thus leading to CHH hypomethylation. In addition, the CHG context at *ONSEN* was significantly lower than average TEs (S5C Fig). In contrast, the CHH context was higher at *ONSEN* (S5C Fig), further indicating that the binding of CMT3 may negatively regulate CMT2-mediated CHH methylation. In the absence of CMT3, the binding of CMT2 on *ONSEN* was indeed slightly increased (Fig 4C) accompanied with increased CHH methylation level and high H3K9me2 level under heat stress (Figs 3 and 5F).

Increased CHH methylation at transposons has been shown to be associated with high temperature accessions of *Arabidopsis thaliana* [30]. In Arabidopsis, heat stress has been reported to trigger upregulation of critical genes involved in the RdDM pathway and induce increased DNA methylation at certain loci, for instance, retrotransposon *AtSN1* and protein-coding genes [31,32]. However, we did not observe any CHH hypermethylation at *ONSEN* under heat stress (Fig 3E). This result may correlate with the limit of siRNA levels that are required to trigger DNA methylation [2], as the 21nt-siRNA level originated from *ONSEN* is very low immediately after heat stress and gradually increases during recovery [12]. Unlike the accessions continuously grown at low or high temperatures, our heat stress experiment was conducted at 37°C for only 24 hours, which may not be sufficient for the DNA methylation change to occur. This suggests that the dampened *ONSEN* activation in the *cmt3* mutant is unlikely due to the effect of heat stress on DNA methylation level. Our study suggests that short period of heat treatment does not affect overall CHH methylation but it is sufficient for H3K9me2 changes to occur at *ONSEN* (Fig 5F). In Arabidopsis, heat stress can cause nucleosome eviction, global rearrangement of the 3D genome, and heterochromatin decondensation without altering DNA methylation [22,27,28] (S9B Fig). H3K9me2 is enriched at heterochromatin and required for the silencing of TEs [33]. Consistently, we found a reduction of H3K9me2 at *ONSEN* under heat (Fig 5F), which is in agreement with decreased proportion of condensed nuclei after heat stress (S4 Fig).

It has been reported that *ONSEN* is initially reactivated by heat-induced transcription factor HSFA2 through recognizing the heat shock element within the LTR [11]. DNA methylation is responsible for suppressing *ONSEN* reactivation after heat stress [12]. On the other hand, TEs have also evolved survival strategies in the host genome to escape transcriptional repression and regulate host responses. For instance, cytosine mutation to thymine of CpG dinucleotide can cause CpG loss and disable the methylation on TE DNA, allowing TEs to take on regulatory functions in the host genome [34]. In terms of transposon survival strategies, *ONSEN* may utilize CMT3 to prevent excessive CHH methylation mediated by CMT2, thus escaping transcriptional suppression by the host plant. The survival of TEs accompanied by transposition in the host genome possibly provides additional regulatory mechanisms for the plant response to severe stress conditions. Future studies on characterizing the consequences of *ONSEN*

transposition in the stress response will shed light on the TE-based regulatory mechanism. In conclusion, distinct from the canonical view that DNA methyltransferases generally act as repressors in the transcriptional regulation of TEs, our findings have uncovered a non-canonical function of the DNA methyltransferase CMT3 in preventing TE silencing under heat stress, specifically for the *ONSEN/ATCOPIA78* family.

## Materials and methods

### Plant materials and heat stress treatment

All *Arabidopsis thaliana* wild-type and mutants were in Columbia (Col-0) ecotype background, except *cmt3-7* (CS6365) in the Landsberg *erecta* (Ler) background. The T-DNA insertion mutant *cmt2-3* (SALK_012874), *cmt3-7* (CS6365), and *cmt3-11* (CS16392) were obtained from the Arabidopsis Biological Resource Center. *kyp-4* (*suvh4*) *suvh5-2 suvh6-1* (*suvh456*) [35] was crossed with *cmt3-11* to generate *cmt3 suvh456* quadruple mutant. The *drm1 drm2* (*drm1/2*), *cmt2 cmt3*, *drm1 drm2 cmt3* (*dd cmt3*), and *drm1 drm2 cmt2 cmt3* (*ddcc*) mutants were gifts from Dr. Steven Jacobsen's lab [4,36,37].

The seeds were surface sterilized with 70% ethanol plus 0.01% Triton X-100 for 15 minutes, washed twice with 100% ethanol, and then sowed onto plates containing half-strength Murashige and Skoog (½ MS) medium and 0.8% agar. The seeds were stratified in the dark at 4˚C for 3 days and then grown under a light intensity of 100 μmol m$^{-2}$ s$^{-1}$ with 24h light at 22˚C. For heat stress, the 10-day-old seedlings were subjected to 37˚C for 24 hours. For time-course heat treatment, 7-day-old seedlings were treated by 37˚C for indicated time and samples were collected at each time-point. Samples were collected immediately after heat stress and frozen in liquid nitrogen.

### Construction of plasmids and generation of transgenic plants

The genomic sequence of CMT2, including 1.36 kb of the promoter, was amplified using genomic Col-0 DNA as a template. A list of primers is in S2 Table. The PCR products were digested by *Sac*I and *Xma*I and then ligated to pCAMBIA1306 with a C-terminal 3x FLAG tag. The plasmid was transformed into the Agrobacterium strain GV3101, and the recombinant strain was transformed into the *cmt2-3* background (SALK_012874) by the floral dip method. The seeds were selected on hygromycin plates (35 mg/L). Homozygous single-insertion lines (with a 3:1 ratio of hygromycin resistant vs. non-resistant in T$_2$) were used. To generate *gCMT2-FLAG* in the *cmt2 cmt3* background, a homozygous line of *gCMT2-FLAG*/*cmt2-3* was crossed with *cmt3-11* (SALK_148381). F$_2$ plants were genotyped for a homozygous *cmt2-3* *cmt3-11* background. Homozygous *gCMT2-FLAG*/*cmt2-3 cmt3-11* F$_3$ plants were screened by hygromycin selection. FLAG-CMT3/*cmt3*, FLAG-CMT3bah3/*cmt3*, FLAG-CMT3chr3/*cmt3*, Myc-CMT3/*cmt3*, and Myc-CMT3cat/*cmt3* have been previously described [3]. The *DRM2-FLAG* transgenic lines have been described [38].

### Chromatin Immunoprecipitation (ChIP)

H3K9me2 and H3 ChIP was performed as previously described [39]. Two grams of 10-day-old non-stress and heat-stress treated Col-0, *cmt3-11*, and *cmt2-3* seedlings were ground into a powder and crosslinked in nuclei isolation buffer (10 mM HEPES pH 8, 1 M sucrose, 5 mM KCl, 5 mM MgCl$_2$, 5 mM EDTA, 0.6% Triton X-100, 0.4 mM PMSF, and protease inhibitor cocktail (Roche; 14696200)) with 1% formaldehyde for 15 min at room temperature. Crosslinking was quenched by adding glycine to a final concentration of 125 mM, and the homogenate was filtered through Miracloth (Millipore, 475855). Nuclei were pelleted by centrifuging

at 4000 rpm for 20 min at 4˚C. The pellet was washed with nuclei isolation buffer II (0.25 M sucrose, 10 mM Tris-HCl, pH 8, 10 mM MgCl$_2$, 1% Triton X-100, 1 mM EDTA, 5 mM β-mercaptoethanol, 0.4 mM PMSF, and protease inhibitor cocktail tablet), then resuspended with 0.3 mL of nuclear lysis buffer (50 mM Tris-HCl pH 8, 10 mM EDTA, 1% SDS, 0.4 mM PMSF, and protease inhibitor cocktail) and kept on ice for 10 min. The lysate was diluted with 1.7 mL of ChIP dilution buffer (1.1% Triton X-100, 1.2 mM EDTA, 16.7 mM Tris-HCl pH 8, 167 mM NaCl, 0.4 mM PMSF, and protease inhibitor cocktail) before shearing the chromatin by sonication using a Covaris S220 focused-ultrasonicator (Covaris). Human H3.2-FLAG-HA chromatin (~300 ng) was added to the supernatant as a spike-in control. This mixture was equally split into two parts for incubating with 5 μg H3 antibody (Abcam, ab1791) and 5 μg H3K9me2 antibody (Abcam, ab1220), respectively, for overnight with constant rotation at 4˚C. The antibodies were preincubated with 25 μL Dynabeads Protein G (for H3K9me2, Invitrogen, 10003D) or Protein A (for H3, Invitrogen, 10001D) for 4–6 hours. After sequential washes with low-salt buffer (150 mM NaCl, 0.1% SDS, 1% Triton X-100, 2 mM EDTA, 20 mM Tris-HCl pH 8), high-salt buffer (500 mM NaCl, 0.1% SDS, 1% Triton X-100, 2 mM EDTA, 20 mM Tris-HCl pH 8), LiCl buffer (0.25 M LiCl, 1% NP-40, 1% sodium deoxycholate, 1 mM EDTA, 10 mM Tris-HCl pH 8), and TE buffer (10 mM Tris-HCl pH 8, 1 mM EDTA), the DNA–protein complex was eluted with elution buffer (1% SDS, 0.1 M NaHCO$_3$) and cross-linking was reversed by adding 200 mM NaCl and incubating at 65˚C for 6 h. After sequential RNase and proteinase K treatment, DNA was purified by the standard phenol-chloroform method.

CMT2-FLAG and CMT3-FLAG ChIP were performed as previously described [25]. Nuclei were isolated from two grams of 10-day-old non-stress, and heat-stress treated seedlings and cross-linked using the same method as described above. After washing with nuclei isolation buffer II, the nuclei were resuspended with MNase buffer (50 mM Tris-HCl, pH 7.5, 50 mM NaCl, 5 mM MgCl$_2$, 5 mM CaCl$_2$, 10% glycerol, 0.1% Nonidet P-40, 0.11 mM PMSF, and protease inhibitor cocktail tablet) and sheared by sonication, followed by MNase digestion for 15 min at 37˚C. EDTA and EGTA were added to 5 mM to stop MNase digestion. Human H3.2-FLAG-HA chromatin (~300 ng) was added to the supernatant as a spike-in control, and incubated with anti-FLAG M2 magnetic beads (Sigma-Aldrich; M8823) overnight with constant rotation at 4˚C. After sequential washes with MNase buffer, high-salt MNase buffer with 300 mM NaCl, LiCl buffer, and TE buffer. The DNA–protein complex was eluted with elution buffer and reverse cross-linked using the same method described above.

## RNA extraction and quantitative RT-PCR

Total RNA was extracted with PureLink RNA mini Kit (Invitrogen; 12183025). 1 μg RNA was treated with DNase I (NEB; M0303) and reverse transcribed into cDNA using ProtoScript II First Strand cDNA Synthesis (NEB; M0368). Quantitative PCR was performed by CFX96 Real-Time System (Bio-Rad) using the SYBR Green Master Mix (Bio-Rad; 1725125) reagent.

## Library construction, sequencing, and data analysis

For RNA sequencing, the extracted RNA was converted to cDNA library with the NEBNext Ultra RNA Library Prep Kit for Illumina (NEB; E7770), NEBNext Poly(A) mRNA Magnetic Isolation Module (NEB; E7490), and NEBNext Multiplex Oligos for Illumina (NEB; E6609). The libraries were sequenced on the NextSeq500 platform (Illumina), and nearly 10 million reads were obtained for each library. Low-quality reads were removed by Trimmomatic version 0.39 [40] using the following parameters "LEADING:20 TRAILING:20 SLIDINGWINDOW:4:20 MINLEN:36". Clean reads were mapped to the Arabidopsis reference genome (TAIR10) using HISAT2 version 2.1.0 with default parameters [41]. Reads were counted by

transcript models. Differentially expressed genes were selected by the adjusted p-value calculated using edgeR version 3.20.9 with default settings [42].

For bisulfite sequencing, raw sequencing data were downloaded from NCBI GEO (GSE39901). Adapter sequence and low-quality reads were trimmed FASTP [43]. Clean reads were aligned to Arabidopsis TAIR10 genome with BSMAP 2.90 using the following parameters "-w100 -v2 -n1 -r1". Methylation at every cytosine was determined by using bsmap's methratio.py script, processing only unique reads and removing duplicates. Differentially methylated regions (DMRs) were detected with MethylKit [44]. The threshold methylation difference for DMRs in each sequence context was adjusted to 40% for mCG, 20% for mCHG, and 10% for mCHH. DMRs were considered significant at q < 0.01. To find DMRs that lied within TEs, lists of DMRs and all TEs from TAIR10 were compared using BEDtools [45] to find overlaps.

For ChIP-seq, raw sequencing data were downloaded from NCBI GEO (GSE51304). Low quality reads were trimmed by Trimmomatic version 0.39 using the following parameters "LEADING:20 TRAILING:20 SLIDINGWINDOW:4:20 MINLEN:25". Clean reads were aligned to Arabidopsis TAIR10 genome with Bowtie2 version 2.4.1 with default parameters [46]. ChIP-seq peaks were called by callpeak function in MACS2 version 2.2.6 [47].

### Bisulfite sanger-sequencing

Genomic DNA was isolated from 10-day-old seedlings with illustra Nucleon Phytopure Genomic DNA Extraction kits (GE Healthcare; RPN8510). Bisulfite conversion and desulfonation were performed using the Methylcode Bisulfite Conversion Kit (Invitrogen; MECOV50), following the manufacturer's protocol. 400 ng genomic DNA were used for each treatment and the following PCR analysis. Primers were listed in S2 Table. Twenty clones were sequenced for each region.

## Supporting information

**S1 Fig. *ONSEN* transcription is reduced in *cmt3* mutant under heat stress.** RT-qPCR showing relative *ONSEN* transcript level in the corresponding mutants under heat stress. All bars represent mean + SD from three biological replicates. The relative *ONSEN* transcripts were first normalized to *18S rRNA*, and then to Col-0. Student's *t*-test, *P < 0.05; **P < 0.01; ***P < 0.001. ns indicates not significant.
(TIF)

**S2 Fig. Similar transcript levels of *ROMANIAT5* and *AtCOPIA37* in Col-0 and *cmt3* under heat stress.** The relative transcript levels were determined by RT-qPCR, first normalized to *ACT7*, and then to Col-0. All bars represent mean + SD from three biological replicates. P value was determined by Student's *t*-test.
(TIF)

**S3 Fig. The expression level of heat stress transcription factor A2 (*HSFA2*) and heat shock protein *HSP90.1* is similar in Col-0 and *cmt3*.** The transcripts were detected by RT-qPCR and normalized to *18S rRNA*. All bars represent mean + SD from four biological replicates. Student's *t*-test, ns indicates not significant. NS, non-stress; HS, heat stress.
(TIF)

**S4 Fig. 4',6-diamidino-2-phenylindole (DAPI) staining of nuclei in Col-0 and *cmt3* mutant.** (A) Representative images of nuclei showing condensed, intermediate, and dispersed chromocenters by DAPI staining. (B) Bar graph indicating the proportion of nuclei displaying condensed, intermediate, and dispersed chromocenters in *cmt3* and Col-0. N indicates the total number of nuclei counted. NS, non-stress; HS, heat stress. ** P < 0.01 by Chi-square test.

ns indicates not significant.
(TIF)

**S5 Fig. Transposons with hyper CHH methylation in *cmt3*.** (A) Bar charts showing the proportion of each transposable element (TE) family in the genome that contains *cmt3* hyper CHH differentially methylated regions. The *P* values were calculated using Fisher's exact test, $^*P < 0.05$, $^{**}P < 0.01$, $^{***}P < 0.001$. (B) Genome browser snapshots of TEs showing hyper CHH methylation in *cmt3*. (C) The percentage of C context (CG, CHG, CHH) in *ONSEN* and other TEs with hyper CHH methylation in *cmt3*. The *P* values were calculated using Fisher's exact test.
(TIF)

**S6 Fig. CMT2 mutation restored compromised *ONSEN* activation in *cmt3* under heat stress.** RT-qPCR showing the relative *ONSEN* transcript level in Col-0, *cmt3-11*, *cmt2-3*, and *cmt2 cmt3* under heat stress. All bars represent mean + SD from three biological replicates. The *ONSEN* transcripts were first normalized to *ACT7*, and then to Col-0. Student's *t*-test, $^{**}P < 0.01$; $^{***}P < 0.001$. ns indicates not significant.
(TIF)

**S7 Fig. The transcript and protein levels of CMT2 are unchanged in *cmt3* mutant relative to Col-0.** (A) RT-qPCR showing relative transcript levels of *DRM2*, *CMT2*, and *CMT3* in Col-0, *drm1 drm2* (*drm1/2*), *cmt2-3*, and *cmt3-11* under heat stress. The relative transcripts were first normalized to *ACT7*, then to Col-0. Data are mean + SD from three biological replicates. Student's *t*-test, $^{**}P < 0.01$. (B) Immunoblot analysis of CMT2 protein in CMT2 tagged lines in *cmt2* (CMT2-FLAG/*cmt2*) and *cmt2 cmt3* background (CMT2-FLAG/*cc*). Arrows indicate CMT2-FLAG protein. Actin serves as a loading control.
(TIF)

**S8 Fig. The protein levels of CMT2, CMT3, and DRM2 under time-course heat stress.** (A and B) The protein levels of CMT2-FLAG in *cmt2* (A) and *cmt2 cmt3* (B) backgrounds after time-course heat treatments. (C and D) The protein levels of CMT3-FLAG (C) and DRM2-FLAG (D) after time-course heat treatments. 7-d-old seedlings were subjected to 37°C heat treatment for indicated time and two biological replicates were present at each time point. Actin and ponceau staining serve as loading controls. (E) Relative transcript levels of *CMT2* and *CMT3* in Col-0 plants treated with 37°C heat for indicated time. The transcript levels were normalized to *ACT7*. Data are mean ± SD from three technical replicates at each time point.
(TIF)

**S9 Fig. H3K9me2 level under heat stress.** (A and B) ChIP-qPCR showing H3K9me2 (A) and H3 (B) abundance at *ONSEN* in Col-0, *cmt3*, and *cmt2* under NS and HS. ChIP samples were first normalized to input, and then to the respective spike-in of human chromatin. *ACT7* served as a negative control. *TA3* was used as a positive control. All bars represent mean + SEM from three biological replicates with all technical replicates shown. Student's t-test, $^*P < 0.05$; $^{**}P < 0.01$; $^{***}P < 0.001$. ns indicates not significant. (C and D) Immunoblots of H3 and H3K9me2 in CMT2-FLAG/*cmt2* (A) and CMT3-FLAG/*cmt3* (B) plants. 7-d-old seedlings were subjected to 37°C treatment for indicated time points. Two biological replicates were present at each time point.
(TIF)

**S10 Fig. The *ONSEN* transcription in time-course of heat stress.** Relative transcript level of *ONSEN* in plants treated with 37°C heat for indicated time. The transcript levels were relative level of *ONSEN* to *ACT7*. 7-d-old seedlings were subjected to 37°C treatment for indicated

time. For each time point, data are mean ± SD from three technical replicates. Student's *t*-tests were performed against Col-0 at the same time point. *P < 0.05; **P < 0.01; ***P < 0.001. (TIF)

**S1 Table. RNA-sequencing statistics.**
(XLSX)

**S2 Table. List of primers used in this study.**
(XLSX)

**S1 Data. RNA seq data.**
(XLSX)

**S2 Data. TEs with *cmt3* hyper CHH DMRs.**
(XLSX)

## Acknowledgments

We would like to thank Dr. Hidetoshi Saze for technical assistance for the whole-genome bisulfite analysis, and Professor Atsushi Kato for his valuable suggestions on experimental techniques and design of the experiment.

## Author Contributions

**Conceptualization:** Hidetaka Ito, Xuehua Zhong.

**Data curation:** Kosuke Nozawa, Jiani Chen, Jianjun Jiang, Sarah M. Leichter, Takamasa Suzuki.

**Formal analysis:** Kosuke Nozawa, Jiani Chen, Jianjun Jiang, Sarah M. Leichter.

**Funding acquisition:** Hidetaka Ito, Xuehua Zhong.

**Investigation:** Kosuke Nozawa, Jiani Chen, Jianjun Jiang, Masataka Yamada, Takamasa Suzuki.

**Project administration:** Hidetaka Ito, Xuehua Zhong.

**Supervision:** Hidetaka Ito, Xuehua Zhong.

**Validation:** Kosuke Nozawa.

**Visualization:** Jiani Chen, Jianjun Jiang.

**Writing – original draft:** Jiani Chen, Jianjun Jiang, Hidetaka Ito, Xuehua Zhong.

**Writing – review & editing:** Jianjun Jiang, Sarah M. Leichter, Fengquan Liu, Hidetaka Ito, Xuehua Zhong.

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
