## [Decision Letter · Decision Letter 0]

2 Apr 2021

Dear Dr Zhong,

Thank you very much for submitting your Research Article entitled 'DNA methyltransferase CHROMOMETHYLASE3 is required for ONSEN transposon activation in heat stress' to PLOS Genetics.

The manuscript was fully evaluated at the editorial level and by independent peer reviewers. The reviewers appreciated the attention to an important problem, but raised some substantial concerns about the current manuscript. Based on the reviews, we will not be able to accept this version of the manuscript, but we would be willing to review a much-revised version. We cannot, of course, promise publication at that time.

Each of the reviewers were interested in the research presented in this work and had positive comments about some aspects of the study.  However, each of the reviewers raised important questions about aspects of the study.  Reviewer 1 points out that an alternative model might explain the observed data.  Reviewer 2 highlights the lack of molecular evidence to support the proposed mechanism.  Reviewer 3 had some questions about why the study was limited to a single ONSEN locus and why some of the molecular work was not conducted on additional genotypes used in this study.  Each of these are important feedback and would need to be productively addressed in a revised version of the manuscript. Should you decide to revise the manuscript for further consideration here, your revisions should address the specific points made by each reviewer. We will also require a detailed list of your responses to the review comments and a description of the changes you have made in the manuscript.  The revised work would likely be evaluated by the same reviewers and would need to satisfy their concerns.

If you decide to revise the manuscript for further consideration at PLOS Genetics, please aim to resubmit within the next 60 days, unless it will take extra time to address the concerns of the reviewers, in which case we would appreciate an expected resubmission date by email to plosgenetics@plos.org.

[LINK]

We are sorry that we cannot be more positive about your manuscript at this stage. Please do not hesitate to contact us if you have any concerns or questions.

Yours sincerely,

Nathan M. Springer

Associate Editor

PLOS Genetics

Wendy Bickmore

Section Editor: Epigenetics

PLOS Genetics

Reviewer's Responses to Questions

**Comments to the Authors:**

Reviewer #1: The study by Nozawa, Chen et al describes an interesting mechanism whereby a DNA methyltransferase thought to be involved in silencing of repeats/transposons is actually needed to prevent silencing. This is an interesting discovery that will impact the way people think about how these pathways function in plant genomes. This study shows that CMT3 is required to prevent ectopic H3K9me2 accumulating at ONSEN upon heat stress. The ectopic H3K9me2 recruits CMT2 to bind and methylate DNA thereby reducing ONSEN activation compared to its activation upon heat stress in wild type. This study nicely incorporates genetics, WGBS and ChIP data to convincingly show that ONSEN requires CMT3 for heat stress induced transcription.

Overall, this study will make a nice contribution to the field. However, I would like the authors to consider a different model that in my opinion better describes the data and fits with recently published studies in the field that are considering the role of heterochromatin homeostasis.

The interpretation of the data is that CMT2 methylating CHH is what leads to silencing in cmt3 mutants. There is no doubt CMT2 is required for this process as nicely shown by the genetic data. This leads the authors to conclude that the function of CMT3 is to prevent binding of CMT2 thereby allowing transcriptional activation upon heat stress. Their data nicely show that CMT3 is a better binder of H3K9me2 regions compared to CMT2, which is what leads to this model. It is certainly a possibility for what is happening.

Another model that should be presented, which I think better describes the data, is the increase in H3K9me2 that occurs at ONSEN in cmt3 is what leads to the dampened transcriptional response. CMT2 binding and methylating CHH is important, but it is secondary to greater amounts of H3K9me2. More H3K9me2 equals more CMT2 and CHH as there is no CMT3 to compete with. I don’t believe CMT3’s function is to compete for binding with CMT2 to enable activation upon heat stress (although it’s certainly one possibility). This is not to say H3K9me2 alone is sufficient for the dampened result, as it would need CMT2 to help maintain a feedback loop. Loss of either would result in greater transcription upon heat stress (as cmt2/3 mutants show in this study).

The observed situation at ONSEN under heat stress fits models whereby cmt3 mutants leads to H3K9me2 redistribution and an imbalance in heterochromatin homeostasis. Recent work by Zhang Y, et al in PNAS present evidence for the idea of heterochromatin homeostasis. This model can also be concluded using the data presented in this study. Therefore, I think the authors should include two potential models 1) the model presented in the study and 2) a model whereby increased H3K9me2 due to redistribution of H3K9m2 as a result of loss of CMT3.

Minor comments:

1. It states state multiple times in the abstract and the main text that cmt3 is required for heat induced expression of ONSEN. As show in Figure 1, ONSEN activation occurs in cmt3 mutants, it’s just not as strong. Therefore CMT3 isn’t require, yet it’s important for a full transcriptional response.

2. How many CHG sites are in the ONSEN locus, specifically where CHH is observed. Is it possible the low CHG is because there are few sites to actually methylate?

3. It was established years ago by Guoll and Baulcombe that CMT2 preferentially catalyzes CWA methylation in Arabidopsis. The study would benefit from specifically measuring CWA levels as opposed to CHH. This study should also be cited as part of this manuscript.

4. It is presented numerous times that CMT3 is activating transcription. I do not agree with this conclusion. This conclusion is made based on a cmt3 mutant. CMT3 is preventing silencing, which is not the same as activating transcription even though both result in the same amount of transcript abundance.

5. Line 104 = do not equate chromatin decondensation with “open chromatin”. Open chromatin refers to accessible chromatin regions bound by transcription factors that are typically 400-600bp in size. This is unrelated to the observed structural decondensation in this study.

6. ChIP seq would be more compelling to show CMT2/CTM3 binding if possible, although this is not required.

Reviewer #2: In this manuscript entitled “DNA methyltransferase CHROMOMETHYLASE3 is required for ONSEN transposon activation in heat stress” the authors investigate the consequences of mutations in CMT2 and CMT3 for the transcriptional activation of the COPIA element ONSEN in response to heat. CMT2 and CMT3 are involved in the maintenance of DNA methylation in heterochromatic regions. Unexpectedly, the authors observed a reduction of ONSEN transcript level in the cmt3 mutant. The authors propose that this is caused by an increase of CMT2-mediated CHH methylation and H3K9me2 at ONSEN. The authors propose that in wild type, CMT3 inhibits CMT2 binding and CHH methylation of ONSEN, and by this function promotes ONSEN transcription under heat.

My major criticism on this manuscript is that the conclusions are based on rather minor changes of DNA methylation and H3K9me2 that have been mainly obtained under non-stressed conditions. However, since the model is based on changes happening under heat stress, changes of epigenetic marks should be investigated under heat stress conditions.

Major comments:

1. The authors propose that CMT3 directly binds to ONSEN and impairs binding of CMT2. However, there is only little CHGm on ONSEN, which seems at odds with this model. It is possible that CMT3 has a larger effect on ONSEN under heat, which could explain this discrepancy. However, data shown in Figure 3D are not convincing and it is also unclear how they were generated. Additional evidence would be required to support the proposed model.

2. Figure 3B and 5B: The authors propose that CHH methylation in ONSEN is increased in cmt3. This is not obvious based on the data presented. Also the metagene plots do not allow to judge quantitative changes. Boxplots would be more suitable to judge whether there are statistically significant changes of CHHm in ONSEN. This would need to be done under non-stress and stress conditions.

3. Figure 5E: The data would need to be normalized to H3 rather than INPUT. Since heat stress causes chromatin decondensation, any changes in H3K9me2 levels could rather be a consequence of nucleosome density changes than changes in H3K9me2 per nucleosome.

4. The authors propose that CMT3 is not sufficient for ONSEN repression, whereas CMT2 is. A discussion on the possible differences of CMT3 and CMT2 to mediate repression is required.

5. Different qPCR experiments were performed with different controls to normalize; is there any justification?

Minor comments:

6. The proposed role of CMT3 in the activation of transposable elements should be revised; the authors implicate that CMT3 has an active role in TE activation; but its role is rather indirect in preventing repression. This should be clarified to avoid misconceptions.

7. L153 “negatively correlating with the increased CHH methylation level”: Rephrase to ”consistent with decreased CHH methylation”.

8. L222-224 rephrase this sentence; it implicates that CMT2 is required for ONSEN activation.

9. Fig1A: The captions indicate two biological replicates, while the barplot shows three datapoints for some groups.

10. Fig2 and Fig 4A,B: Single data points should be added to the barplots as done for Fig 1A

11. Fig 5D: The statistical comparisons need to be specified, especially the double asterisks over the bar of suvh456 and cmt3 suvh456, do they refer to the comparison to Col-0?

12. Line 299-306: References for all mutants should be included.

Reviewer #3: The authors show an absence of ONSEN upregulation upon heat shock in cmt3 mutant, together with an increase of H3K9me2 compared to WT. They provide data showing that both CM2 and CMT3 bind an ONSEN locus, leading to a model of ONSEN regulation.

The findings are interesting although they are restricted to ONSEN family. Why is it so? The authors mention in the discussion that other TEs could be regulated this way but they didn't find any in their RNAseq data.

The manuscript is a bit confusing as most experiments are done on one ONSEN locus (what about the others?) and with few replicates. The manuscript would benefit from having data on transgenic lines (Fig 4) and mutants (Fig 5C, D) under stress conditions.

In the discussion part the authors mention that the HS is too short to induce DNA methylation changes, however H3K9me2 marks are drastically reduced n WT (Figure 5E). Did the authors anchored the primers for ChIP in the flanking borders of ONSEN? The very high level of eccDNA (not bearing histone marks) could affect the measure.

Fig 1A, Fig 2, Fig 5D,E: I wonder if it is feasible to calculate SD with only two biological replicates.

Fig 4: why were these experiments done without heat shock?

Line 36: class II TEs do change copy number when they jump before replication that's why they are abundant in genomes.

Line 43: evolutionary new insertions, do you mean 'recent insertions'?

Line 114-115: correct typo

Line 135: showed a greatly increase, please correct

Line 136: Sanger not sanger

Line 174: please correct the sentence

Line 208: a unique ONSEN, please correct to unique family

Line 240: what is the link with the Swedish accessions?

Line 263: the short HS treatment performed in this study is not sufficient for DNA methylation changes to occur but it is sufficient for H3K9me2 changes to occur (see Fig 5E), this should be notified here.

Line 276: reproduction?

Figure 4: on this Figure the authors show that both CMT3 and CMT2 bind an ONSEN locus in WT, without HS. What happens during the HS?

Figure 6: the two panels on the right (cmt3 mutant) are similar but with different outcomes for transcription, the H3K9 methyltransferase activity should be shown here. Also the grey/white arrows in the middle are confusing (between WT and mutant).

**Have all data underlying the figures and results presented in the manuscript been provided?**

Reviewer #1: Yes

Reviewer #2: Yes

Reviewer #3: Yes

PLOS authors have the option to publish the peer review history of their article (what does this mean?). If published, this will include your full peer review and any attached files.

Reviewer #1: No

Reviewer #2: No

Reviewer #3: No

---

## [Decision Letter · Decision Letter 1]

11 Jun 2021

Dear Dr Zhong,

Thank you very much for submitting your Research Article entitled 'DNA methyltransferase CHROMOMETHYLASE3 is required for full ONSEN transposon activation in heat stress' to PLOS Genetics.

The manuscript was fully evaluated at the editorial level and by independent peer reviewers. The reviewers appreciated the attention to an important topic but identified some concerns that we ask you address in a revised manuscript.  Reviewer 1 had a relatively simple request regarding presentation of data for a specific sequence context for some analyses.  Reviewer 2 makes an important point about the proper normalization for the ChIP data and potential issues with the model and interpretations in the title or abstract.

We therefore ask you to modify the manuscript according to the review recommendations. Your revisions should address the specific points made by each reviewer.

[LINK]

Yours sincerely,

Nathan M. Springer

Associate Editor

PLOS Genetics

Wendy Bickmore

Section Editor: Epigenetics

PLOS Genetics

Reviewer's Responses to Questions

**Comments to the Authors:**

Reviewer #1: The new manuscript is improved for clarity and will make a nice contribution to the field. My only remaining comment is to consider presenting CWA vs non-CWA in Figure 3/5 D. If this is entirely a CMT2 dependent process the CHH shown in 3d/5d will likely be mostly CWA. i think that would be good for the reader to know.

Reviewer #2: In the revised version of the manuscript, the authors included new data showing CMT2 binding and H3K9me2 accumulation under heat stress. This is an important improvement. Based on their model, CMT3 prevents CMT2 binding, causing decreased ONSEN expression in cmt3 under heat stress. The data of non-heat treated cmt3 mutants are consistent with the model. However, the new data reveal that there is no detectable CMT2 binding in cmt3 under heat, which the authors explain by showing that CMT2 is unstable under heat. What remains inconsistent with the model is the fact that H3K9me2 levels on ONSEN are not increased in cmt3 under heat (Figure 5F). One possible explanation could be that the ChIP data have been normalized to INPUT, which, as pointed out before, will lead to a misinterpretation of the data since the chromatin decondenses under heat. Most likely, the level of H3K9me2 per nucleosome will not change under heat, contrary to what the data show. The Western blots are not addressing this point since nucleosome changes in response to heat include gain and loss of nucleosomes. Preferentially this point should be experimentally addressed (ChIP normalized to H3); but at least it needs to be discussed that the data need to be treated with caution since normalization to input does not account for changes in nucleosome density.

The title remains misleading; the fact that in the cmt3 mutant ONSEN is less expressed, does not mean that CMT3 is required for ONSEN expression. The correct description of the data, which should be reflected in the title, is that "CMT2 prevents activation of ONSEN under heat stress", or "CMT3 prevents silencing of ONSEN under heat stress". To propose that CMT3 activates ONSEN would require to show that by artificially targeting CMT3 to the ONSEN locus leads to ONSEN activation, which has not been tested and seems also unlikely.

The abstract should be rephrased along the same lines.

**Have all data underlying the figures and results presented in the manuscript been provided?**

Reviewer #1: Yes

Reviewer #2: Yes

PLOS authors have the option to publish the peer review history of their article (what does this mean?). If published, this will include your full peer review and any attached files.

Reviewer #1: No

Reviewer #2: No

---

## [Editor Report · Decision Letter 2]

12 Jul 2021

Dear Dr Zhong,

We are pleased to inform you that your manuscript entitled "DNA methyltransferase CHROMOMETHYLASE3 prevents ONSEN transposon silencing under heat stress" has been editorially accepted for publication in PLOS Genetics. Congratulations!

Yours sincerely,

Nathan M. Springer

Associate Editor

PLOS Genetics

Wendy Bickmore

Section Editor: Epigenetics

PLOS Genetics

Comments from the reviewers (if applicable):

**Data Deposition**

http://datadryad.org/submit?journalID=pgenetics&manu=PGENETICS-D-21-00236R2

**Press Queries**

---

## [Editor Report · Acceptance letter]

27 Jul 2021

PGENETICS-D-21-00236R2 

DNA methyltransferase CHROMOMETHYLASE3 prevents ONSEN transposon silencing under heat stress 

Dear Dr Zhong, 

We are pleased to inform you that your manuscript entitled "DNA methyltransferase CHROMOMETHYLASE3 prevents ONSEN transposon silencing under heat stress" has been formally accepted for publication in PLOS Genetics! Your manuscript is now with our production department and you will be notified of the publication date in due course.

With kind regards,

Katalin Szabo

PLOS Genetics

On behalf of:
